# Genome-Wide Identification and Functional Analysis of *C2H2* Zinc Finger Transcription Factor Genes in the Intertidal Macroalga *Pyropia haitanensis*

**DOI:** 10.3390/ijms26094042

**Published:** 2025-04-24

**Authors:** Jiajia Xie, Dehua Ji, Yan Xu, Kai Xu, Chaotian Xie, Wenlei Wang

**Affiliations:** 1Fisheries College, Jimei University, Xiamen 361021, China; 2State Key Laboratory of Mariculture Breeding (Jimei University), Ningde 352100, China; 3Key Laboratory of Healthy Mariculture for the East China Sea, Ministry of Agriculture and Rural Affairs, Xiamen 361021, China

**Keywords:** intertidal adaptability, *Pyropia haitanensis*, C2H2 type zinc finger protein, whole-genome analysis, Dapseq

## Abstract

The possible regulatory effects of C2H2 zinc finger proteins, which are important transcription factors, on intertidal seaweed responses to abiotic stress are unclear. This study was conducted to comprehensively analyze the *C2H2* gene family of a representative intertidal seaweed species (*Pyropia haitanensis*) and clarify its genomic characteristics and biological functions. A total of 107 *PhC2H2* zinc finger protein-encoding genes distributed on five *P. haitanensis* chromosomes were identified and divided into three subgroups. The expression levels of 85, 61, 58, 45, and 41 *PhC2H2* genes responded in the maturation of filaments, high-temperature, salt, low-irradiance, and dehydration stress, respectively. The *PhC2H2* gene family was conserved during *Porphyra* evolution, with no indications of large-scale genome-wide replication events. On average, *PhC2H2* genes had more transposable element (TE) insertions than *Pyropia yezoensis* and *Porphyra umbilicalis*
*C2H2* genes, suggesting that TE insertions may have been the main driver of *PhC2H2* gene family expansion. A key gene (*PhC2H2.94*) screened following a quantitative trait locus analysis was significantly responsive to high-temperature stress and was associated with photosynthesis, peroxisomes, the ubiquitin proteasome pathway, and the endoplasmic reticulum-related protein processing pathway, which contribute to the stress tolerance of *P. haitanensis*. Additionally, *PhC2H2.94* transgenic *Chlamydomonas reinhardtii* exhibited increased tolerance to heat stress. This study provides new insights and genetic resources for characterizing the molecular mechanism underlying intertidal seaweed responses to abiotic stresses and breeding stress-resistant macroalgae.

## 1. Introduction

Red algae (Rhodophyta) are ancient eukaryotic algal species that are still distributed worldwide and serve as an important component of marine ecosystems, with an important position in the origin and evolution of species [1,2]. Red algae generally grow in extreme environments, such as intertidal zones that are substantially affected by tides. They are exposed to various abiotic stresses, such as dehydration, high temperatures, intense radiation, and high salinity, thereby requiring a unique adaptive mechanism [3,4,5,6]. Therefore, elucidating the adaptive evolution of intertidal organisms, including red algae, is of paramount importance. *Pyropia haitanensis*, which is a representative intertidal red algal species, is highly resistant to stress. For example, it can survive even after losing 95% of its water content [7]. Moreover, *P. haitanensis* is rich in nutrients and is an economically valuable algal species. Its production in 2023 reached approximately 162,000 tons, accounting for about 75% of the total *Pyropia*/*Porphyra* production in China [8]. Thus, clarifying the stress resistance mechanism of *P. haitanensis* may provide a theoretical basis for characterizing the adaptive evolution of intertidal red algae, while also providing theoretical guidance for breeding stress-resistant macroalgae.

According to comparative genomic analyses, *Pyropia*/*Porphyra* species adapted to the intertidal zone at least 738.5 million years ago [9], with *P. haitanensis* adapting to intertidal conditions in various ways [10,11]. In *P. haitanensis*, the perception of external stress induces signal transduction through Ca^2+^, mitogen-activated protein kinase cascade phosphorylation, and phosphatidylinositol, which synergistically activate downstream anti-stress genes or pathways [12]. For example, *PhCaM1* interacts with *DnaJ* to regulate the high-temperature shock system and regulates DNA repair and stress-resistant protein synthesis through ribosomal proteins. It can also interact with anion channel proteins to regulate Ca^2+^ and energy entry and exit between mitochondria and cytoplasm [13]. Furthermore, through inhibiting protein synthesis, recruiting molecular chaperones, and activating the ubiquitin-proteasome system, it helps to remove misfolded proteins and reduce the harmful effects of endoplasmic reticulum stress [14,15]. Hence, the *P. haitanensis* response to stress in the intertidal zone is mediated by a systematic, effective, and unique adaptive mechanism involving a series of complex physiological and metabolic changes. However, previous studies on stress resistance mechanisms have mainly focused on metabolic pathways and key genes. Accordingly, the mechanisms regulating pathways and genes related to the adaptive evolution of *Pyropia*/*Porphyra* remain unclear.

Transcriptional regulation involves cis-acting regulatory elements, such as promoters and enhancers, and trans-acting factors, including transcription factors (TF) that control the transcription of different genes [16]. Zinc finger proteins (ZFPs), which are one of the most abundant TFs in eukaryotes, are ubiquitous in animals, plants, and fungi [17]. In the classical zinc finger structure, the binding center Zn^2+^ is combined with histidine (His) and/or cysteine (Cys) repeats and folded in various arrangements via hydrophobic interactions to form a stable ‘finger’ configuration [18]. *C2H2*-type ZFPs TFs, which are the most studied ZFPs [19], are important for plant transcriptional regulation, RNA metabolism, and other processes [20,21]. In 1992, *Petunia hybrida EPF1* was the first *C2H2*-ZFP gene to be cloned; the encoded TF may be involved in activating the expression of the 5-enolpyruvylshikimate-3-phosphate synthase-encoding gene (*EPSPS*), thereby regulating *P. hybrida* development [22]. Additional C2H2-ZFPs have gradually been identified and analyzed in other plants. For example, 122, 211, 118, and 321 *C2H2*-ZFP-encoding genes were identified in *foxtail millet* [23], maize [24], tobacco [25], and soybean [26], respectively. Related research on C2H2-ZFPs has demonstrated that C2H2-ZFPs contribute to important mechanisms associated with plant life cycles and stress responses [27,28,29]. For example, Bai et al. identified a key *C2H2* TF gene (*OSIC1*) that is activated by salt, drought, polyethylene glycol 6000, and abscisic acid, with the encoded TF directly up-regulating the expression of a cupric polyamine oxidase gene (*PalCuAOζ*) in vitro to enhance H_2_O_2_ accumulation in guard cells, thereby regulating stomatal closure in response to osmotic stress [30]. For example, Xu et al. identified 102 *RrC2H2* genes in *Rosa rugosa*, and revealed that the heterologous expression of *RrC2H2-8* in *Arabidopsis thaliana* can significantly improve its salt tolerance [31]. Zhou et al. identified and analyzed the whole genome of the C2H2 zinc finger protein family in *Opisthopappus* plants. The results showed that under salt stress, the expression of some genes changed significantly under salt stress [32]. Li et al. analyzed the genome-wide characteristics of the C2H2-type zinc finger protein family in six representative terrestrial plants from mosses to angiosperms, revealing their potential functions and evolutionary relationships in dehydration and rehydration reactions [33].

In addition, the TF family emphasized in related studies on algae includes C2H2 zinc finger proteins. Xu et al. studied the genome diversity and niche differentiation of *Bathycoccus*, a marine eukaryotic phytoplankton, and found that the *C2H2* zinc finger protein gene family was enriched in branches (BI and BIV) adapted to cold waters, indicating that the C2H2 zinc finger protein is related to its cold adaptation mechanism [34]. Petroll et al. analyzed transcription-associated proteins (TAP) and found that the copy number of the *C2H2* gene family in *Porphyridium purpureum* was more similar to the copy number in multicellular red algae. It is speculated that the *C2H2* gene was acquired after the differentiation of Cyanidiales and other lineages, and the abundance of the *C2H2* zinc finger transcription factor family may be related to the acquisition of morphological complexity [35]. Thus, reports published to date on plant C2H2-ZFPs have confirmed that C2H2-ZFPs play a key regulatory role in many biological processes. However, their regulatory effects on the responses of intertidal macroalgae to abiotic stress are still unclear.

In this study, 107 *C2H2*-ZFP genes in *P. haitanensis* were identified and clustered. Moreover, their phylogenetic relationships, gene structures, conserved motifs, chromosomal locations, and promoter cis-acting elements underwent a bioinformatics analysis. We determined that *PhC2H2* expression profiles vary depending on the type of stress and the developmental stage. More specifically, the expression levels of *PhC2H2* family members were mainly up-regulated in the late filamentous maturation stage and following an exposure to salt stress or high-temperature stress. On the basis of these findings and the results of quantitative trait locus (QTL) and quantitative real-time polymerase chain reaction (qRT-PCR) analyses, a high-temperature-responsive gene (*PhC2H2.94*) was screened for sequence analyses and target gene identification via cloning and DAP-seq analysis. The study results may provide a theoretical basis for future investigations on the effects of high-temperature stress on *P. haitanensis C2H2*-ZFP family genes, while also providing new insights and genetic resources for further clarifying the molecular basis of macroalgal responses to abiotic stress as well as the breeding of stress-resistant macroalgae.

## 2. Results

### 2.1. Identification of Transcription Factors and Phylogenetic Analysis of C2H2 Gene Family

TF-encoding genes from nine species (seven representative red algae, *A. thaliana*, and *O. sativa*) were clustered, and the copy numbers of 87 TF genes were determined (Appendix A). *C2H2* copy numbers were highest in red algae, especially *P. haitanensis*, which had 107 gene copies. In contrast, *A. thaliana* had 100 *C2H2* gene copies. Notably, *O. sativa* lacked *C2H2* genes. To explore the evolutionary relationships among *C2H2* genes in red algae, a phylogenetic tree was constructed based on 554 C2H2 protein sequences from eight species (Figure 1). *C2H2* genes were divided into three subgroups (A, B, and C). The number of *C2H2* genes from different species varied among the subgroups. Subgroup A, which was the smallest, included genes from *Porphyridium purpureum*, *Galdieria sulphuraria*, *A. thaliana*, and *Gracilariopsis chorda*. Subgroup B included *C2H2* genes from all species, but most were from *A. thaliana*. Subgroup C, which was the largest, mainly included *C2H2* genes from *P. haitanensis*, *P. yezoensis*, and *Porphyra umbilicalis* as well as some *C2H2* genes from *P. purpureum*, *A. thaliana*, *Chondrus crispus*, and *G. chorda*. After considering plant evolutionary processes, we selected six representative chlorophytes, two bryophytes, two charophytes, and two phaeophyceae and constructed an evolutionary tree comprising 21 species (Figure 2A). *C2H2* copy numbers were highest in bryophytes, followed by rhodophytes and chlorophytes.

### 2.2. Physicochemical Properties and Predicted Cis-Acting Elements of PhC2H2 Genes

Based on an analysis of physicochemical properties, the 107 identified *P. haitanensis* C2H2-ZFPs consisted of 91–2037 amino acids, with a molecular weight of 9.90–19.92 kDa and a theoretical pI of 5.42–12.25. Instability coefficients were greater than 40, indicating that these proteins were unstable. *C2H2* gene and coding sequence (CDS) lengths were 506–21,343 and 276–6114 bp, respectively. Subcellular localization results indicated 97% of the *C2H2* genes encoded proteins located in the nucleus. The predicted *PhC2H2* promoter cis-acting elements (Figure 2B) provided insights into gene functions. Specifically, we identified cis-acting elements related to plant hormones (e.g., 39.6% MeJA-responsive elements, 28.4% abscisic acid-responsive elements, 0.47% auxin-responsive elements, 0.44% salicylic acid-responsive elements, and 0.22% gibberellin-responsive elements) and abiotic stress responses (e.g., 28.7% light-responsive elements, 1.9% low-temperature-responsive elements, and 0.18% defense-related and stress-responsive elements) (Appendix A).

### 2.3. Structures, Motifs, and Transposons of PhC2H2 Genes

To examine *PhC2H2* structural characteristics, the motifs, domains, and CDS (Figure 2C) of *PhC2H2* genes were analyzed. The results showed that *PhC2H2* structures differed between groups. Most of the genes contained motifs 1, 3, 5, and 10, which are typical conserved motifs (CX2CX3FX5LX2HX3H) of *C2H2* TF genes. In addition, motif 6 may be a leucine-rich region (L-box) that is important for protein–protein interactions. A total of 34 domains were predicted in *PhC2H2* gene family members, including zinc finger domains and some superfamily domains. Almost all *PhC2H2* genes contained the ZF-C2H2 domain.

The introns of *C2H2* genes in seven red algae were analyzed. Compared with the genes in the other examined species, the genes in *Porphyra* species had (on average) more introns, especially *P. haitanensis* (average of 1.9 introns per *PhC2H2* gene), and longer introns. Notably, *C2H2* genes from *G. sulphuraria*, which grows in extreme environments, were second only to *P. haitanensis C2H2* genes in terms of the number of introns, with an average of 1.5 introns per *C2H2* gene. To explain the multiple copies of *PhC2H2* genes, we identified and counted transposable element (TE) insertions in these genes. A total of 347 TE insertion sites were detected in 73 *PhC2H2* genes, with an average of 4.8 TE insertions per *PhC2H2* gene. Moreover, the average number of TE insertions was higher for Subgroup C genes than for Subgroup A/B genes (Appendix A).

### 2.4. Chromosomal Localization, Replication, and Collinearity of PhC2H2 Genes

*PhC2H2* genes were mapped to chromosomes to reveal their chromosomal distribution. Chromosome 1 (i.e., longest chromosome) had 26 *PhC2H2* genes, chromosome 2 had 25 *PhC2H2* genes, chromosome 3 had 23 *PhC2H2* genes, chromosome 4 had 20 *PhC2H2* genes, and chromosome 5 had 13 *PhC2H2* genes. According to an analysis of *PhC2H2* gene replication, there were no detectable segmental duplications, but there was a tandem duplication in the middle of chromosome 4 (Figure 3A). Furthermore, an examination of *PhC2H2* genes on different *P. haitanensis* chromosomes revealed a lack of collinearity, with the exception of the limited collinearity detected for the genes on chromosome 1 (Figure 3B). In addition, collinearity was detected among *P. haitanensis*, *P. yezoensis*, and *P. umbilicalis C2H2* genes (Figure 3C).

### 2.5. Gene Expression Pattern Analysis and qRT-PCR Verification

Transcriptome data were used to analyze *PhC2H2* expression patterns (Figure 4A). A total of 85 *PhC2H2* genes were differentially expressed during the maturation of filaments, 63 *PhC2H2* genes were responsive to high-temperature stress, 60 *PhC2H2* genes were responsive to salt stress, 44 *PhC2H2* genes were responsive to low-irradiance stress, and 40 *PhC2H2* genes were responsive to dehydration stress. Among these genes, 24 were responsive to different stresses. Moreover, the expression patterns of genes differed among developmental stages and stress conditions (Figure 4B). The expression levels of four *C2H2*-ZFP-encoding genes annotated in previously identified high-temperature-resistance-related QTLs in *P. haitanensis* [36] were verified by qRT-PCR (Appendix A; Figure 4C). Of these genes, *PhC2H2.94* was more highly expressed in the high-temperature-resistant strain than in the high-temperature-sensitive strain. During the high-temperature stress treatment, *PhC2H2.94* expression was rapidly up-regulated in the high-temperature-resistant strain at 15 min (5.1 times higher than the control expression level). In contrast, in the high-temperature-sensitive strain, significantly up-regulated *PhC2H2.94* expression was detected at 3 h (5 times higher than the control expression level). In terms of the overall trend, *PhC2H2.94* expression was initially up-regulated by high-temperature stress, but then gradually decreased to a stable level. Our bioinformatics analysis showed that *PhC2H2.94* has two ZnF-C2H2 domains and a typical conserved motif comprising an α-helix and an anti-parallel β-sheet (Appendix A).

### 2.6. DAP-seq

Following a DAP-seq analysis, 2845 peaks were identified, with a total length of 1,269,941 bp and an average length of 446 bp. The identified peaks were concentrated near the transcription start site (TSS), especially 400 bp downstream of TSS (Figure 5A). The distribution of peaks in gene functional elements was as follows: 39.96% in promoters, 25.31% in 5′ untranslated regions (UTRs), 14.9% in 3′ UTRs, 12.33% in introns, 4.6% in exons, 1.86% in intergenic regions, and 0.95% in downstream regions (Figure 5B). Additionally, *PhC2H2.94* was able to bind to conserved motifs, including CT(G/C)TA, (A/G)(G/A)A(C/G)A, and CA(C/T)(C/A)C (Figure 5C). The enriched KEGG metabolic pathways among the peak-related genes were RNA splicing, photosynthesis, oxidative phosphorylation, ubiquitination-mediated proteolysis, and protein synthesis and processing in the endoplasmic reticulum (Figure 5D).

### 2.7. Stress Resistance of Transgenic C. reinhardtii Expressing PhC2H2.94

The results showed that the survival state of transgenic *PhC2H2.94* was significantly better than that of wild type under high-temperature stress, and the *PhC2H2.94* gene expression level in transgenic *C. reinhardtii* was significantly increased after 3 h of high-temperature stress. These results indicated that the expression of the *PhC2H2.94* gene in transgenic *C. reinhardtii* was induced by high-temperature stress, and the transfer of the *PhC2H2.94* gene improved the high-temperature tolerance of *C. reinhardtii*. In addition, stress-related genes such as antioxidase coding genes (*CAT*, *SOD*) and heat shock protein coding genes (*HSP70A*, *HSP90A*) in transgenic and wild-type *C. reinhardtii* at different time points under high-temperature stress were verified by qRT-PCR. These related genes in gene expression under high-temperature stress were significantly higher. (Appendix A).

## 3. Discussion

Transcriptional regulation is critical for plant responses to abiotic stress, with TFs playing an important regulatory role in abiotic stress responses and other life-cycle-related activities. Therefore, this study identified TF genes in the *P. haitanensis* genome. Notably, the *C2H2* gene copy number (107) was higher than the copy numbers of the other TF genes in *P. haitanensis* as well as the copy numbers of *C2H2* genes in other red algae (8–79) and *A. thaliana* (100) (Appendix A). The number of gene families in different plant species is different, reflecting the trend of diversification in the process of evolution. According to the evolution of plant species and the change of copy number of the *C2H2* zinc finger protein gene family, it is found that except for 72 copies in *P. purpureum*, the copy number of *C2H2* family members in multicellular red algae is significantly higher than that in single-cell red algae. The results of this study are consistent with previous research [35]. It is speculated that the C2H2 zinc finger protein may appear earlier in the process of evolution. In addition, the growth of *Pyropia*/*Porphyra* in the intertidal zone experienced repeated changes in flood and low tides, while moss is a representative species of the transition from aquatic to terrestrial environment. It is speculated that the expansion of the C2H2 zinc finger protein family in *Pyropia*/*Porphyra* and moss may be related to the adaptation of plants to more complex environments [37]. It is speculated that the high copy number of the C2H2 zinc finger protein family in *P. haitanensis* may be related to the adaptability of the intertidal zone.

According to the classification in the model organism *A. thaliana*, *C2H2*-ZFP-encoding genes can be divided into Subgroups A, B, and C [38]. Subgroup C, which is the largest, may represent an ancestral evolutionary branch. In this study, C2H2 zinc finger protein was divided into three groups (Group A, B, C), with the largest number in Group C, which mainly included the C2H2 zinc finger protein gene of *P. yezoensis*, *P. umbilicalis*, and *P. haitanensis*, as well as some C2H2 zinc finger protein genes of *P. purpure*, *A. thaliana*, *C. crispus*, and *G. chorda*. Compared with higher plants, red algae have a smaller genome, with no signs of large-scale genome-wide replication events [39]. The compact genome structure and relatively few introns in most genes in red algae are the result of an ‘evolutionary bottleneck’ [40,41]. Our analysis also failed to detect large-scale genome-wide replication events in *P. haitanensis*, which differs from the related findings for plants [42,43] and green algae [44]. Earlier studies showed that TE insertions contribute to the expansion of gene families, which may enhance species adaptability and diversity [45]. For example, in *A. thaliana*, TE insertions are enriched near genes related to responses to environmental stimuli [46]. TE insertions were also an important driver of CAM evolution; the related multi-copy genes have many TE insertions and are differentially expressed [47]. In an earlier study, we determined that although TEs account for only 13.07% of the *P. haitanensis* genome, they played an important role in the evolution of gene functions [36]. In the present study, the average number of TE insertions was greater in *P. haitanensis* genes (4.8) than in *P. yezoensis* (2.5) and *P. umbilicalis* (2.5) genes. Compared with the genes in the other subgroups, Subgroup C genes had higher copy numbers and more TE insertions, suggesting that TE insertions may be the main driving force for the expansion of the *PhC2H2* gene family.

In the *PhC2H2* gene family, some gene structures and motifs were conserved, while others underwent diversification. Compared with the corresponding genes in the other *P. yezoensis* and *P. umbilicalis*, *PhC2H2* genes had more and longer introns as well as higher copy numbers, which may lead to more alternative splicing and mutations that result in functional diversification [48,49]. According to an analysis of their promoter cis-acting elements, *PhC2H2* genes may be involved in plant hormone pathways and abiotic stress responses. Moreover, *PhC2H2* genes may primarily affect the late stage of filament development and maturation. The expression of these genes is significantly induced under abiotic stress conditions (e.g., high-temperature, salinity, and dehydration), implying that C2H2-ZFPs play a key role in *P. haitanensis* growth and development and adaptations to environmental stressors. Additionally, the 107 *PhC2H2* gene family members are evenly distributed on the five *P. haitanensis* chromosomes, suggesting that the encoded TFs broadly regulate gene expression. The results of the gene replication event score showed that the *C2H2* gene family was conserved during the evolution of *Pyropia*/*Porphyra* species. The limited collinearity and gene replication in *Pyropia*/*Porphyra* species suggest that large-scale whole-genome replication events likely did not occur in *Pyropia*/*Porphyra* genomes. In summary, genes encoding C2H2-type ZFPs were conserved during the evolution of *Pyropia*/*Porphyra* species, with no evidence of a large-scale whole-genome replication event. Diverse gene structures and multiple copies of *PhC2H2* family members may be conducive to regulating the expression of various stress-resistance-related genes under harsh and variable intertidal environmental conditions.

Due to global warming, *P. haitanensis* is often subjected to high-temperature-induced damage, which seriously affects the sustainable development of the *P. haitanensis* industry. In order to further elucidate the pathways and genes regulated by C2H2-ZFPs in *P. haitanensis*, we screened four candidate genes encoding C2H2-ZFPs based on high-temperature QTL data (Appendix A). The results showed that the expression levels of *PhC2H2.94*, *PhC2H2.284*, and *PhC2H2.291* genes in high-temperature-resistant lines were higher than those in high-temperature-sensitive lines, and they responded quickly and positively to high-temperature stress at 15 min. The difference was that the *PhC2H2.73* gene showed a continuous downward trend, which seemed to play a negative regulatory role. Also, the expression levels of 85, 61, 58, 45, and 41 *PhC2H2* genes responded in the maturation of filaments, high-temperature, salt, low-irradiance, and dehydration stress, respectively. This suggests this gene family underwent subfunctionalization after expansion. The C2H2-ZFP subfamily has acquired significant heterogeneity in the C-terminal region and expression patterns, exhibiting dynamic patterns of subfunctionalization and neofunctionalization, thereby promoting environmental adaptation and speciation [50].

The bioinformatics analysis of *PhC2H2.94* indicated that the gene was a typical *C2H2* zinc finger transcription factor [51] (Appendix A). According to DAP-seq results, *PhC2H2.94* may bind to different motifs, including CT(G/C)TA. Most of the zinc finger motifs in C2H2-ZFPs differ, indicating that they likely bind to different DNA sequences, enabling the regulation of different genes. For *P. hybrida ZPT2-2*, which contains two conserved zinc finger domains, the best binding sequences for the N-terminal and C-terminal zinc fingers are AGC(T) and CAGT, respectively [52]. In *A. thaliana*, *ZPT2*-related proteins reportedly bind to A(G/C)T repeats [53]. Target genes regulated by this TF were mainly associated with RNA splicing, oxidative phosphorylation, photosynthesis, peroxisomes, UPP, and endoplasmic reticulum-related protein processing. These processes are closely related to *P. haitanensis* stress resistance. For example, under high-temperature stress conditions, *P. haitanensis* decreases energy consumption due to light-induced damage and carbon assimilation, while also decreasing photosynthetic activities to prevent excessive ROS production, activating the antioxidant system to maintain the redox balance, and activating the HSP system and UPP to maintain intracellular protein synthesis, folding, and scavenging [12,54,55].

At present, the gene editing and molecular verification methods of macroalgae are gradually developing, but the whole process is still in its primary stage [56]. The construction of the genetic system of *Pyropia*/*Porphyra* still faces many challenges, such as the complexity of the genome, the low efficiency of genetic transformation, the limitations of the selection marker system, and the difficulties in life cycle and reproductive characteristics [57]. *Pyropia*/*Porphyra* has not yet achieved gene function verification in vivo, and *C. reinhardtii* can be used as a good carrier for gene function verification of *Pyropia*/*Porphyra* [58]. We generated *PhC2H2.94*-expressing transgenic *C. reinhardtii*. The results showed that *PhC2H2.94* was significantly induced in *C. reinhardtii* under high-temperature treatment, and compared with wild-type *C. reinhardtii*, the transgenic *C. reinhardtii* significantly improved its high-temperature tolerance. In addition, stress-related genes such as catalase (*CAT*), superoxide dismutase coding gene (*SOD*), and heat shock proteins HSPs (*HSP70A*, *HSP90A*) in transgenic and wild-type *C. reinhardtii* at different time points under high-temperature stress were verified by qRT-PCR. These confirmed stress-resistant-related genes in gene expression under high-temperature stress were significantly higher, and transgenic type *C. reinhardtii* gene expression of the whole genome is higher than the wild type, suggesting that algae may go through *PhC2H2.94* gene regulation of the antioxidant system, heat stress protein synthesis stress-resistance-related pathways, and then respond to the *P. haitanensis* process of high-temperature stress. The above results of our transgene and DAP-seq study preliminarily analyzed the response mechanism of the key transcription factor *PhC2H2.94* in stress. Recently, Wang et al. cloned for the first time the PolIII type promoter of *Neopyropia yezoensis* to guide the expression of RNA, and realized the precise knockout of targeted genes of *Pyropia*/*Porphyra* based on the CRISPR/Cas system, obtaining a series of gene-edited algae strains with breeding significance [59]. We hope that in the future, we will build genetic systems such as CRISPR/Cas or RNAi to achieve more direct verification, and apply them to the selection of resistant strains.

In summary, this study for the first time conducted a whole-genome analysis and functional verification of the important transcription factor C2H2 zinc finger protein family in macroalgae *P. haitanensis*, an ideal species for studying intertidal adaptability, and explored its role and mechanism in the growth, development, and stress response of *P. haitanensis*, which is conducive to the analysis of gene structure and biological function, the mining of key genes, and the analysis of plant regulatory information network.

## 4. Materials and Methods

### 4.1. Identification and Phylogenetic Analysis of C2H2 Gene Family Members

For a *C2H2* family homologous gene cluster analysis, genome data sets for 21 species were downloaded from NCBI (https://www.ncbi.nlm.nih.gov/, accessed on 27 April 2023) and Ensembl Plants (http://plants.ensembl.org/index.html, accessed on 27 April 2023), whereas genomic data for *P. yezoensis* and *P. haitanensis* were generated in our laboratory. Diamond and OrthoMCL were used to identify homologous genes between species (Appendix A). According to the sequence alignment results generated by Diamond, gene pairs with an E value < 1 × 10^−5^ and a query coverage of 30% were considered to be homologous genes between species. OrthoMCL was used to classify homologous genes in the same family. C2H2 protein sequences for each species were extracted and downloaded. ClustalW in MEGA-X was used for the multiple sequence alignment of C2H2 proteins, after which the multiple sequence alignment results were quickly pruned. The neighbor-joining method was used to calculate phylogenetic distances 1000 times for the construction of a phylogenetic tree of *C2H2* gene families from different species. The phylogenetic tree was visualized using iTOL (https://itol.embl.de/, accessed on 25 July 2023).

### 4.2. Structures, Motifs, Cis-Acting Elements, and Physicochemical Properties of PhC2H2 Genes

Conserved motifs were identified using MEME (http://meme-suite.org/tools/meme, accessed on 4 August 2023) and annotated using Pfam (https://pfam-legacy.xfam.org/, accessed on 4 August 2023) and InterProScan (https://www.ebi.ac.uk/interpro/, accessed on 4 August 2023) databases. Protein sequences encoded by *PhC2H2* family members were used as queries to screen the NCBI database to identify conserved domains. *PhC2H2* gene locations and exon/intron information were obtained from available *P. haitanensis* genome information. In addition, 2000 bp promoter regions upstream of the initiation codon (ATG) were extracted from *P. haitanensis* genome information and analyzed using PlantCARE (http://bioinformatics.psb.ugent.be/webtools/plantcare/html, accessed on 11 August 2023) to identify cis-acting elements. TBtools v2.083 was used to visualize gene structures, motifs, and promoter cis-acting elements. ExPasy online tools (http://web.expasy.org/protparam/, accessed on 22 August 2023) were used to calculate the molecular weight, instability coefficient, and isoelectric point (pI) of proteins. WoLF PSORT (http://wolfpsort.org/, accessed on 22 August 2023) was used to predict the subcellular localization of *PhC2H2*.

### 4.3. Analyses of the Chromosomal Localization, Duplication, Collinearity, and Transposons of PhC2H2 Genes

GTF/GFF of TBtools v2.083 was used to visualize the chromosomal positions of *PhC2H2* genes. A collinearity file of *PhC2H2* genes was extracted and processed into a links format for an analysis of repeating fragments. A tandem file was used for an analysis of tandem repeats. Intraspecies and interspecies collinearity analyses of *P. haitanensis* were performed using BLAST and MCScanX plug-ins (accessed on 25 August 2023). The data of repetitive components previously obtained by this laboratory based on homologues and de novo strategy identification were analyzed and statistically analyzed for TE insertion [36].

### 4.4. Treatment of Experimental Materials

*P. haitanensis* WO14-1 (sensitive to high-temperature stress) and WO18-2 (resistant to high-temperature stress) were obtained from the Laboratory of Germplasm Improvements and Applications of *P. haitanensis* at Jimei University. The experimental strains were cultured in Provasoli’ s enrichment solution at 21 ± 0.5 °C, the light intensity was 50~60 μmoL/(m^2^·s), and the photoperiod was 12 L:12 D. The solution was refreshed every 2 days. Cultured algae (15 ± 2 cm long) that were growing well were selected for subsequent analyses. For the high-temperature stress treatment, algae were placed in a constant-temperature incubator set at 32 °C and cultured for 0, 15 min, 30 min, 1 h, 3 h, 6 h, 12 h, and 24 h. Four biological replicates were prepared for each treatment. All samples were immediately frozen in liquid nitrogen and stored at −80 °C. The *Chlamydomonas reinhardtii* strain used in this study was selected from the cell wall deletion strain ‘CC-400 CW15 MT+’ cultured in our laboratory. The control group was cultured in a 150 mL conical flask with shaking at 100 rpm and a temperature of 25 ± 0.5 °C under a light intensity of 50 μmoL/(m^2^·s) and a 14 h light/10 h dark photoperiod. High-temperature treatment conditions were the same as those used for *P. haitanensis*.

### 4.5. Isolation and Purification of Total RNA and Synthesis of cDNA

Total RNA was extracted from *P. haitanensis* samples using an EZNA plant RNA extraction kit (Omega, Cambridge, MA, USA). The purity and quantity of the extracted RNA were initially estimated from OD_260_ and OD_280_ values measured with a Cary50 UV spectrophotometer (Varian, Palo Alto, CA, USA), with RNA integrity and quality subsequently confirmed by 1% agarose gel electrophoresis. The extracted RNA was then reverse transcribed into cDNA using a PrimeScript RT Reagent kit (Takara, Kusatsu, Japan).

### 4.6. Gene Expression Pattern Analysis and Verification by qRT-PCR

Using transcriptome data generated in our laboratory (for different filament developmental stages [60] and following an exposure to high temperature [61], salt [62], low-irradiance [63], or dehydration stress [64]), a high-temperature map of *PhC2H2* expression was constructed using TBtools v2.083. Additionally, the expression levels of four annotated *C2H2*-ZFP-encoding genes in QTL associated with *P. haitanensis* thallus high-temperature tolerance were verified by qRT-PCR. The corresponding gene sequences were extracted from the genome, and qRT-PCR forward and reverse primers were designed using Primer Premier 5 (Appendix A). The amplification efficiency of the primers was 95–105%. *PhUBC* [65] and *β-tubulin* [66] were used as reference genes with *P. haitanensis* and *C. reinhardtii* for the qRT-PCR amplification, respectively, which were performed using the Step One Plus fluorescence quantitative PCR instrument (ABI, Los Angeles, CA, USA). The qRT-PCR program was as follows: 95 °C for 30 s, 40 cycles of 95 °C for 5 s, and 58.6 °C for 30 s. GraphPad Prism 9.5 was used to map the analyzed data. A one-way ANOVA method was used to compare the differences among data groups, with *p* < 0.05 and *p* < 0.01 set as the thresholds for significant and extremely significant differences, respectively.

### 4.7. Cloning and DAP-seq of PhC2H2.94

According to the designed specific primers of *PhC2H2.94*, the full-length clone was amplified by PCR using ApexHF HS DNA polymerase premix kit (AG, Changsha, China) and Mastercycler X50s PCR instrument (Eppendorf, Hamburg, Germany), and the single target fragment with the correct size was recovered and tested. The target gene and HaloTag expression vector were subjected to double-enzyme digestion by adding EcoR V and Hind III homologous arms at both ends of the target gene primer sequence, respectively. The conjugated products were transformed into *Escherichia coli* HST80 receptive cells by heat shock transformation, and positive monoclonal colonies were selected for colony PCR verification and plasmid extraction. *P. haitanensis* thalloid DNA was extracted by the CTAB method to construct a library. Additionally, the TnT SP6 high-yield wheat germ protein expression system was used to express the *PhC2H2.94*–HaloTag fusion protein, which was purified with HaloTag magnetic beads, washed, and filtered to remove non-specific or unconnected protein. After removing DNA that did not bind to TF proteins, the remaining TF-bound DNA fragments were recovered and amplified for a DAP-seq analysis [67]. A genome-wide peak analysis was performed using the obtained DAP-seq read data. Peaks were scanned using MACS2 2.1.2 (q < 0.05). Information regarding peak locations was used to screen for peak-related genes, which were annotated using the ChIPseeker R package 1.16.1. MEME and DREME were used to identify sequence motifs.

### 4.8. Construction of a PhC2H2.94 Expression Vector and Transformation of Chlamydomonas reinhardtii

Referring to the specific steps of gene cloning, *PhC2H2.94* PCR product and expression vector pChlamy_3 were double-digested with Kpn I and Pst I endonuclide (Vazyme, Nanjing, China), respectively. At 50 °C for 10 min, the target gene and expression vector were connected (OK Clon mix Kit, AG, Changsha, China), and the connected products were transformed into *Escherichia coli* DH5α receptor cells. Positive monoclonal colonies were verified by PCR and propagated by culturing. The pChlamy_3 expression vector containing *PhC2H2.94* was purified from confirmed transformants. The target gene was transferred to *C. reinhardtii* cells via glass bead transformation [68], resulting in the successful cloning of *PhC2H2.94* (Appendix A). Subsequently, three wild-type and transgenic *C. reinhardtii* samples were randomly selected for high-temperature stress (32 °C) treatments for 0, 1, 2, and 3 days for apparent and OD_750_ determination, as well as qRT-PCR verification after treatment for 0, 1, 3, and 6 h.

## 5. Conclusions

A whole-genome analysis of 107 *PhC2H2* genes in *P. haitanensis* revealed the relatively broad chromosomal distribution and functional diversity of the *PhC2H2* gene family. Examinations of cis-acting elements and expression patterns indicated that *PhC2H2* gene family members have diverse effects on *P. haitanensis* growth and development and abiotic stress responses. Analyses of gene duplication events, collinearity, and transposons showed that the *PhC2H2* gene family was relatively conserved during *Porphyra* evolution, with TE insertions potentially serving as the primary driving force for the expansion of the *PhC2H2* gene family. A classical *C2H2* zinc finger TF-encoding gene (*PhC2H2.94*) was selected from the QTL candidate gene induced by high temperatures. Its regulatory region was concentrated in the promoter and 5′ UTR. The encoded TF appears to regulate *P. haitanensis* biological processes closely associated with stress resistance (e.g., oxidative phosphorylation, photosynthesis, peroxisomes, UPP, and protein processing via the endoplasmic reticulum) by binding to different conserved motifs (e.g., CT(G/C)TA). In addition, the heterologous expression of *PhC2H2.94* in *C. reinhardtii* was significantly induced by high-temperature stress and promoted *C. reinhardtii* high-temperature tolerance. The results of this study not only promote the functional research of the *P. haitanensis* transcription factor family, providing new ideas for understanding the molecular mechanism by which large seaweeds respond to abiotic stress, but also offer key genetic resources for the breeding of stress-resistant strains.

## Figures and Tables

**Figure 1 ijms-26-04042-f001:**
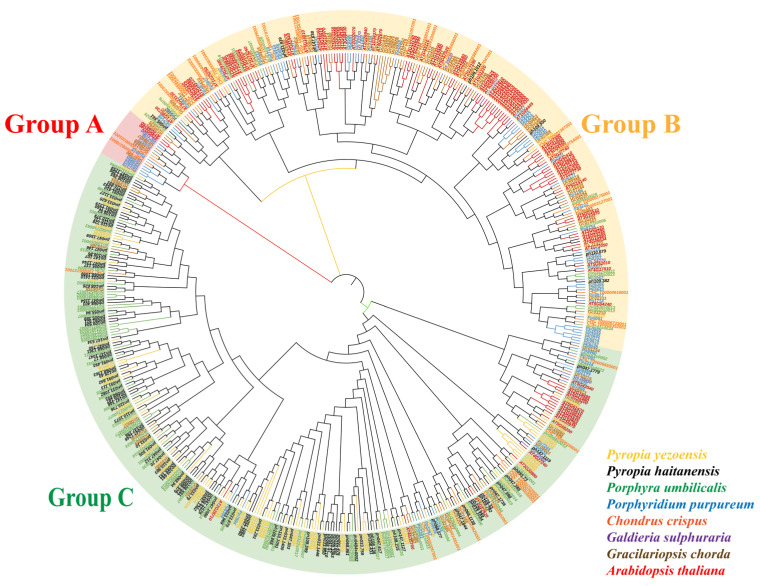
The phylogenetic tree was constructed according to C2H2 zinc finger protein sequences in seven species. Markers with different colors represent different species, whereas differentially shaded areas represent different groups.

**Figure 2 ijms-26-04042-f002:**
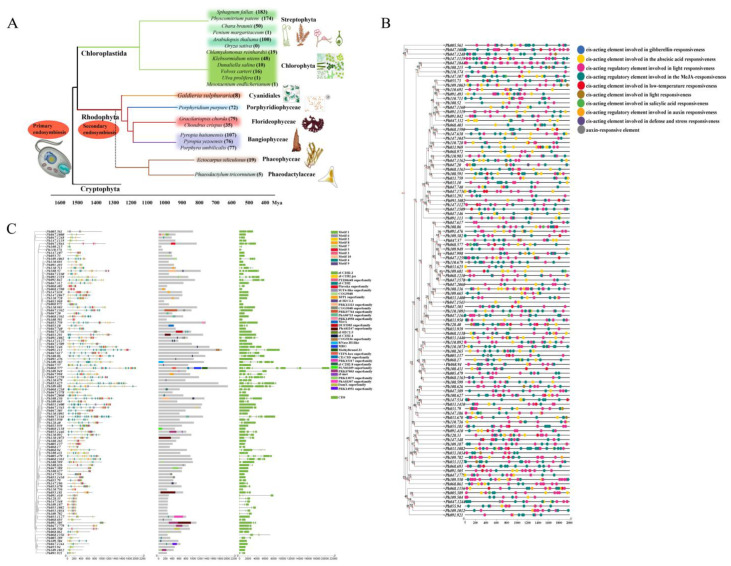
Analysis of the C2H2 gene family evolutionary tree, gene structure, and cis-acting elements. (**A**) Evolutionary tree comprising 21 species, with the number of extracted *C2H2* gene families indicated. (**B**) Cis-acting elements in *PhC2H2* gene family members. (**C**) Ten motifs, 32 domains, and gene structures. Markers with different colors represent different conserved motifs, domains, and CDS at the corresponding positions.

**Figure 3 ijms-26-04042-f003:**
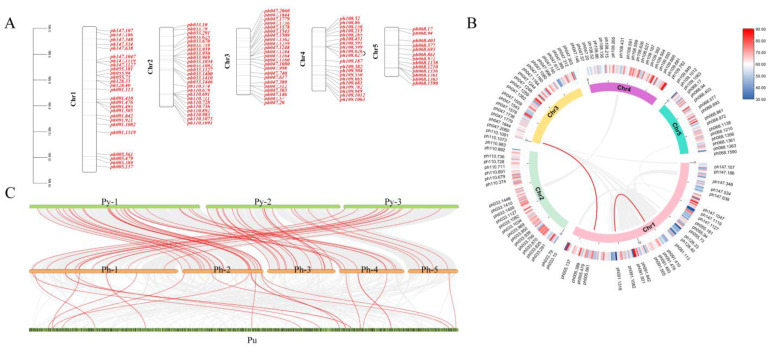
Evolutionary analysis of the *PhC2H2* gene family: (**A**) *PhC2H2* chromosomal localization and replication events. (**B**) Collinearity within *P. haitanensis*. Inner to outer circles represent collinear gene pairs, different chromosomes, gene density, and distribution of *PhC2H2* genes. (**C**) Collinearity between *P. haitanensis C2H2* genes and *C2H2* genes from *P. yezoensis* and *P. umbilicalis*. Collinear gene pairs are indicated by a red line.

**Figure 4 ijms-26-04042-f004:**
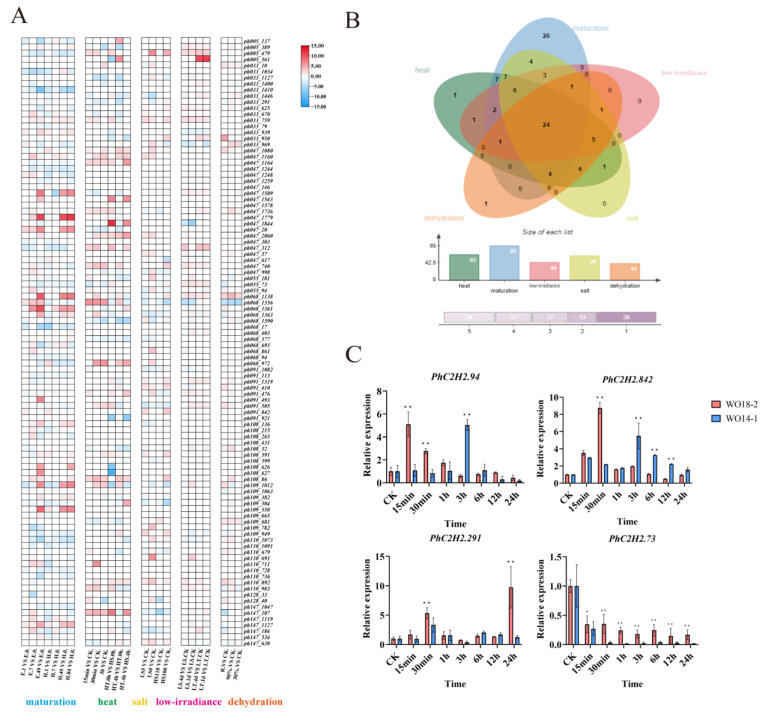
*PhC2H2* expression profiles: (**A**) *P. haitanensis PhC2H2* expression profiles at different filament developmental stages and in response to diverse stresses. For the *PhC2H2* expression profiles in different filament developmental stages, E represents the strain that matures easily, whereas H represents the strain that does not mature easily. The number represents the treatment time. For the *PhC2H2* expression profiles in response to high-temperature treatments, HT represents the heat-resistant strain, whereas HS represents the heat-sensitive strain. For the *PhC2H2* expression profiles in response to salt treatments, LS represents the low-salinity treatment, whereas HS represents the high-salinity treatment. For the *PhC2H2* expression profiles in response to low-irradiance treatments, LS represents the light-sensitive strain, whereas LT represents the light-resistant strain. The percentage indicates the degree of dehydration. (**B**) Venn diagram of *PhC2H2* genes responsive to different stress treatments. (**C**) Expression levels and trends of four QTL candidate genes in different *P. haitanensis* strains under high-temperature stress for different durations. Asterisks indicate significant differences between treatments (* *p* < 0.05, ** *p* < 0.01).

**Figure 5 ijms-26-04042-f005:**
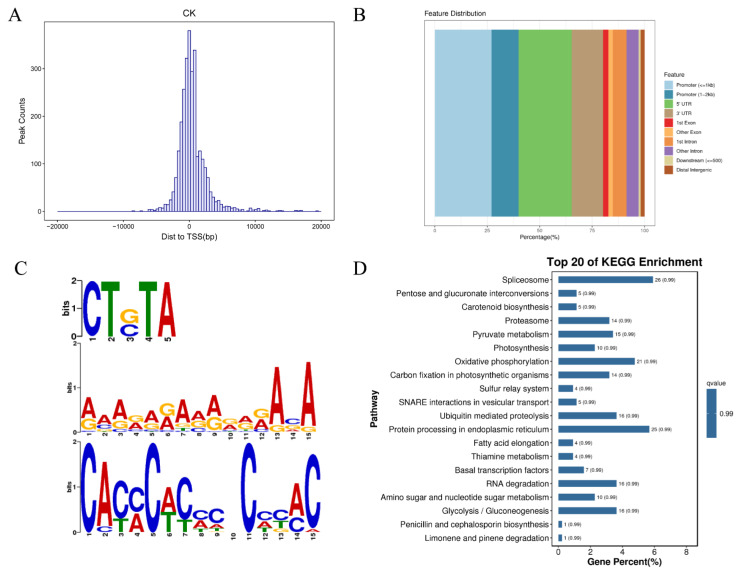
DAP-seq analysis results: (**A**) Peak distribution relative to gene positions. The *y*-axis presents the distance between the peak and the transcription start site (TSS), whereas the *x*-axis presents the number of peaks. (**B**) Distribution of peaks in gene functional elements. (**C**) Significant motifs in the peak sequence. Different colors represent different base types. The height of each letter represents how conserved a particular base is at a particular site (i.e., a tall letter represents a highly conserved base). (**D**) KEGG enrichment cycle diagram; B: KEGG enriched bar chart.

## Data Availability

The original contributions presented in this study are included in the article/Appendix A. Further inquiries can be directed to the corresponding author(s).

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
