# Peer review of "Genome-Wide Identification and Functional Analysis of C2H2 Zinc Finger Transcription Factor Genes in the Intertidal Macroalga Pyropia haitanensis"

_ijms, 2025, doi:10.3390/ijms26094042_

Round 1
Reviewer 1 Report
Comments and Suggestions for Authors
The manuscript is a well-executed study that provides important insights into C2H2 TFs in P. haitanensis. The authors should clarify the novelty of their findings, improve the discussion on functional validation, and provide additional methodological details.
Is the validation of PhC2H2.94 in Chlamydomonas reinhardtii sufficient to support its proposed function in Pyropia haitanensis?
The study suggests that transposable elements (TEs) played a role in expanding the C2H2 gene family. Could additional analyses (e.g., TE insertion age estimation, and functional impact of TEs) confirm this hypothesis? How does the C2H2 gene expansion in P. haitanensis compare with similar expansions in other macroalgae or terrestrial plants?
Are there specific evolutionary pressures that might have driven the high copy number of C2H2 TFs in P. haitanensis? Could a more detailed phylogenetic analysis, including divergence time estimation, provide deeper evolutionary insights?
The study highlights the role of C2H2 TFs in stress adaptation. Could additional physiological or biochemical assays (e.g., ROS quantification, stress marker analysis) further validate these findings? How do the observed expression patterns compare with known stress-responsive genes in P. haitanensis or other macroalgae?
Can the findings be translated into applied research, such as breeding stress-tolerant macroalgae for aquaculture? Are there potential genetic engineering or synthetic biology approaches that could leverage these insights for industrial or environmental applications?
Author Response
Dear Editors and Reviewers:
We would like to thank the editors and reviewer's work devoted to our manuscript and we are very grateful for your valuable suggestions.
According to the comments put forward by the reviewers, we have taken seriously and responded to them. In addition, we have improved the content and provided more necessary information. We have fully addressed each concern and hope that this revised manuscript is now acceptable. Each concern is discussed in detail below.
Reviewer 1Comment 1
The manuscript is a well-executed study that provides important insights into C2H2 TFs in P. haitanensis. The authors should clarify the novelty of their findings, improve the discussion on functional validation, and provide additional methodological details.
Response 1: Thank you for your positive and valuable comments, which certainly help us to improve the quality of our manuscript. We supplemented our innovation points in the discussion section and enhanced more content regarding transgenic verification in the results. For example, in our previous studies, it has been confirmed that the antioxidant system and protein folding pathways play important roles in the stress resistance response of Pyropia haitanensis. This study verified the significant changes of PhC2H2.94 and related stress resistance genes under high-temperature stress through heterologous expression. In addition, we have added more details about the materials and methods, such as the strains and culture conditions of Chlamydomonas reinhardtii, the isolation and purification of total RNA and the synthesis of cDNA, as well as more detailed information related to gene cloning and analysis.
Comment 2
Is the validation of PhC2H2.94 in Chlamydomonas reinhardtii sufficient to support its proposed function in Pyropia haitanensis?
Response 2: Thank you for raising this critical question. We fully understand your concerns regarding functional validation through heterologous expression in C. reinhardtii. CRISPR functionality has been confirmed in two brown algae species (Ectocarpus species 7 and Saccharina japonica), two red algae species (Gracilariopsis lemaneiformis and P. yezoensis), and one green algae species (Ulva prolifera), however, these studies are limited to proof-of-concept demonstrations, and the editing efficiency is relatively low [1, 2]. Unlike seaweed species, C. reinhardtii is a unicellular green alga broadly used for elucidating fundamental biological processes. Many heterologous genes have been expressed in C. reinhardtii, including genes from Pyropia [3-5], higher plants [6], and humans [7]. Therefore, in this study, the successful heterologous expression of PhC2H2.94, a key gene of P. haitanensis, in the model organism C. reinhardtii may provide a basis for further understanding of the function of these genes in P. haitanensis. We provide more discussion in the revised manuscript.
Comment 3
The study suggests that transposable elements (TEs) played a role in expanding the C2H2 gene family. Could additional analyses (e.g., TE insertion age estimation, and functional impact of TEs) confirm this hypothesis? How does the C2H2 gene expansion in P. haitanensis compare with similar expansions in other macroalgae or terrestrial plants?
Response 3: Thank you for your professional advice. In fact, we have tried to analyze the time of TE insertion in the PhC2H2 gene family, but there may be errors in the estimate of the nucleotide replacement rate, which affects the accuracy of the TE insertion time. As shown in Figure 1, we calculated the insertion time of LTR in the P. haitanensis genome and found that the insertion time of LTR in P. haitanensis genome is mainly concentrated in 0.5 - 1 MYA, which may be related to the dynamic changes and adaptive evolution of its genome [8]. And we also consulted relevant literature to provide explanatory references. As shown in Figure 2, it was mentioned in the genome study of the red algae Chondrus crispus that the insertion time of its TEs was concentrated in the last 300,000 years, and its TE expansion may be related to its adaptation to the marine environment [9].
Figure 1. The insertion time of LTR in the genome of P. haitanensis.
Figure 2. Proposed scenario for the evolution of red algae.
In addition, we analyzed the role of TE insertion on the function of the PhC2H2 gene family. For example, as shown in Figure 3 and Table 1, we can find that the quantity trend of TE insertion is roughly consistent with the expression trend of PhC2H2 gene under high temperature stress. When there is TE insertion in genes, genes tend to show more positive expression response, suggesting that TE insertion may have an impact on the response expression of PhC2H2 gene family to stress. In addition, we also analyzed the influence of TE insertion based on the results of systematic evolution. As shown in Table 2, the C2H2 zinc finger protein family can be divided into three groups, among which Group C of Pyropia/Porphyra has the largest number of C2H2 zinc finger protein family members and the highest average TE insertion number. Among them, the number of P. haitanensis is the largest, and its average TE insertion number is also the largest. It is suggested that TE insertion plays a role in the expansion of gene families. Compared with the gene family, gene amplification is mainly caused by whole genome duplication in higher plants [10]. However, compared with terrestrial plants, the genome of red algae is smaller and there is no large-scale genome-wide replication event [11], which is consistent with the results of this study. In the genomic study of P. haitanensis, although TE insertions accounted for only 13.07% of P. haitanensis, they played an important role of gene expression and function [8], and TE insertions contributed to the expansion of gene families [12], which may improve species adaptability and diversity.
Table 1. TE insertion quantity and expression of PhC2H2 gene family under high temperature stress
|
Gene family |
expressed |
unexpressed |
|
67 genes have TE insertions. |
35 |
32 |
|
33 genes had no TE insertion. |
12 |
21 |
Table 2. TE insertion and phylogenetic grouping of PhC2H2 gene family
|
Pyropia/Porphyra |
Group A |
Group B |
Group C |
||||
|
Gene number |
TE insertion number |
Average TE number |
Gene number |
TE insertion number |
Average TE number |
||
|
PH |
0 |
7 |
22 |
3.14 |
100 |
325 |
3.25 |
|
PY |
0 |
5 |
1 |
0.2 |
71 |
82 |
1.15 |
|
PU |
0 |
14 |
9 |
0.69 |
64 |
60 |
0.93 |
Figure 3. Heatmap of TE and transcription under high temperature stress of PhC2H2 Gene family.
Comment 4
Are there specific evolutionary pressures that might have driven the high copy number of C2H2 TFs in P. haitanensis? Could a more detailed phylogenetic analysis, including divergence time estimation, provide deeper evolutionary insights?
Response 4: Thank you for this thought-provoking question, we agree with your ideas and direction of thinking. C2H2 TFs shows significant gene family expansion in many species and is widely involved in stress response, developmental regulation, and epigenetic modification. Genome-wide replication (WGD) or tandem duplication may be the direct cause of the increase in copy number. However, in this study, we found no large-scale genome-wide replication events in the PhC2H2 gene family, only a tandem repeat sequence was found. P. haitanensis as intertidal algae, periodically exposed to extreme environments. The high copy number may be driven by evolutionary pressure, and TE insertion also may contribute to the high copy number. However, when we analyzed the selection pressure of the PhC2H2 gene family, no positive selection genes were found. Perhaps because the selection pressure of the gene family itself is weak, only certain genes in a specific lineage experience positive selection, and the whole gene family level cannot be detected. It is also possible that we are using insufficient analytical methods and reference models.
In addition, a more detailed phylogenetic analysis would also be of great help to the elaboration of our paper. However, we hope you can understand that the quality of our genome assembly is insufficient, there are problems such as microbial contamination, high GC, and high heterozygosity, with BUSCO accounting for 75.5%. There are still many genes that have not been fully annotated or not annotated correctly, and indeed some analyses are limited. We have added this discussion and outlook in this paper. The main purpose and key innovation of this study is the combination of genome-wide analysis of gene families and further screening of key transcription factors to explain their mechanisms of action in response to stress. In future studies, we will expand our research on genome analysis to get a deep and thorough understanding of this problem. Thank you very much for your creativity.
Comment 5
The study highlights the role of C2H2 TFs in stress adaptation. Could additional physiological or biochemical assays (e.g., ROS quantification, stress marker analysis) further validate these findings? How do the observed expression patterns compare with known stress-responsive genes in P. haitanensis or other macroalgae?
Response 5: We agree to your proposal, Our previous research found that under high temperature stress, the expression level of PhGPX gene, one of the important members of the ROS clearance enzyme system in P. haitanensis, showed a trend of up-regulation and then down-regulation [13], and also proved that in response to high temperature stress in P. haitanensis, the ROS clearance pathway was first activated, and then with the persistence of high temperature stress, the ROS clearance ability was weakened [14], which was consistent with the expression trend of PhC2H2 under high temperature stress studied in this study. And we determined the expression of genes encoding antioxidant enzymes and heat shock proteins. As shown in Fig.2, under high temperature stress, the expression levels of stress-related genes (CAT, SOD, HSP70A, HSP90A) in transgenic C. reinhardtii were higher than wild type, indicating that the algae may regulate the antioxidant system, heat stress protein synthesis and other stress-related pathways through PhC2H2.94 gene, and then respond to the high temperature stress process of P. haitanensis.
Figure 2. Relative transcription levels of antioxidant enzymes and heat shock protein genes of transgenic (T2) and wild-type (WT) C.reinhardtii under high temperature stress. * : significant difference, P < 0.05; * * : means extremely significant difference, P < 0.01.
We found that the response of P. haitanensis to high temperature stress can be divided into two stages. First, the algae have a stress response in the early stage of stress, reducing photosynthesis to reduce the production of reactive oxygen species, and reducing unnecessary energy consumption. Afterwards, recombination was performed at the transcriptional level to further activate stress-resistant pathways such as photosynthesis, energy metabolism, and antioxidant systems to resist long-term high temperature stress [15], Huang et al. showed that when the external stress of plants is alleviated, downstream stress-related proteins that bind to transcription factors convert transcription factors back to their original inactive form to inhibit transcriptional activity, thus weakening stress response [16]. For example, Zhang et al. found that the expression of PhbZIP2 transcription factor of P. haitanensis increased significantly at first under high temperature treatment for 5 - 30min, then gradually decreased, and then increased again at 3 - 6h [17], so as to explain the trend that PhC2H2.94 increases first, then decreases, then increases slightly and finally becomes stable.
Comment 6
Can the findings be translated into applied research, such as breeding stress-tolerant macroalgae for aquaculture? Are there potential genetic engineering or synthetic biology approaches that could leverage these insights for industrial or environmental applications?
Response 6: Thank you for your good idea. Relevant studies have confirmed that C2H2 zinc finger protein, as an important family of transcription factors, plays a key regulatory role in various plant biological processes, which is of great significance for identifying and utilizing key genes, analyzing plant stress resistance mechanism, developing stress resistance strains, and improving yield [18].The results of this study showed that members of the PhC2H2 gene family were involved in the response of P. haitanensis to stress such as high temperature, high salt and low salt, drought, and filamentous maturation, which provided a theoretical basis and gene resources for the improvement of stress resistance by using the C2H2 gene family. For example, through CRISPR/Cas system, target genes in laver were accurately knocked out and gene-edited algae strains with breeding significance were obtained [19]. That's what we've been working towards. In summary, the functional study of C2H2 gene family not only provides an important basis for understanding the resistance mechanism of plants and macroalgae, but also provides a certain idea support for the development of genetic engineering in the future, and has a broad application prospect.
Thanks again, the hard work and valuable suggestions of the editors and reviewers are of great help to our improvement in the manuscript. We sincerely hope that the revised article can meet the requirements of International journal of molecular science and be accepted.
References:
[1] De Saeger J, Coulembier Vandelannoote E, Lee H, et al. Genome editing in macroalgae: advances and challenges. Frontiers in Genome Editing, 2024, 6, 1380682.
[2] Wang H, Xie X, Gu W, et al. Gene editing of economic macroalga Neopyropia yezoensis (Rhodophyta) will promote its development into a model species of marine algae[J]. New Phytologist, 2024, 244: 1687-1691.
[3] Kim E, Park HS, Jung Yi, et al. Identification of the high-temperatureresponse genes from Porphyra seriata (Rhodophyta) ESTs and en-hancement of heat tolerance of Chlamvdomonas (Chlorophyta) by expression of the Porphyra HTR2 gene. J Phycol, 2011, 47:821-828.
[4] Chang J, Shi J, Lin J, et al. Molecular mechanism underlying Pyropia haitanensis PhHsp22-mediated increase in the high-temperature tolerance of Chlamydomonas reinhardtii. Journal of Applied Phycology, 2021, 33, 1137-1148.
[5] Wang W, Lin J, Chang J, et al. A RING type ubiquitin ligase PhCUL4 is involved in thermotolerance of Pyropia haitanensis. Algal Research, 2021, 59, 102448.
[6] Siripornadulsil S, Traina S,Verma DPS, et al. Molecular mechanisms of proline-mediated tolerance to toxic heavy metals intransgenic microalgae. The Plant Cell 2002, 14: 2837-2847.
[7] Rasala BA, Mayfield SP. The microalga Chlamydomonas reinhardtii as a platform for the production of human protein ther-apeutics. Bioeng Bugs, 2011, 2: 50-54.
[8] Wang W, Ge Q, Wen J, et al. Horizontal gene transfer and symbiotic microorganisms regulate the adaptive evolution of intertidal algae, Porphyra sense lato[J]. Communications Biology, 2024, (7): 1-13.
[9] Collén, J.; Porcel, B.; Carré, W.; Ball, S.; Chaparro, C.; Tonon, T.; Barbeyron, T.; Michel, G.; Noel, B.; Valentin, K.; et al. Genome structure and metabolic features in the red seaweed Chondrus crispus shed light on evolution of the Archaeplastida. Proc. Natl. Acad. Sci. USA 2013, 110, 5247–5252.
[10] Jiao Z. C., Wang L. P., Du H., et al. Genome-wide study of C2H2 zinc finger gene family in Medicago truncatula[J]. BMC Plant Biology, 2020, 20(1).
[11] Cao T J. Genomics of Bangia fuscopurpurea [D]. Nanjing: Nanjing University, 2019.
[12] Zhang Y, Li Z, Liu J, et al. Transposable elements orchestrate subgenome-convergent and -divergent transcription in common wheat[J]. Nature Communications, 2022, 13(1): 6940.
[13] Zhang H, Xu Y, Ji D, et al. Cloning and expression characteristics of glutathione peroxidase gene from Pyropia haitanensis[J]. Chinese Journal of Aquatic Sciences, 2016, 23(04): 791-799.
[14] Xiao H. Preliminary study on antioxidant enzyme system of Pyropia haitanensis under high temperature and water loss stress [D]. Xiamen: Jimei University, 2014.
[15] Wang W., Teng F, Lin Y., et al. Transcriptomic study to understand thermal adaptation in a high temperature-tolerant strain of Pyropia haitanensis[J]. Plos One, 2018, 13(4): e195842.
[16] Huang Y, Niu C, Yang C, et al. The heat-stress factor HSFA6b connects ABA signaling and ABA-mediated heat responses[J]. Plant Physiology, 2016, 172(2): 1182-1199.
[17] Huang R., Jiang S., Dai M., et al. Zinc ffnger transcription factor MtZPT2-2 negatively regulates salt tolerance in Medicago truncatula[J]. Plant Physiology, 2023,00: 1–14.
[18] Bai Q, Niu Z, Chen Q, et al. The C2H2‐type zinc finger transcription factor OSIC1 positively regulates stomatal closure under osmotic stress in poplar[J]. Plant Biotechnology Journal, 2023, 21(5): 943-960.
[19] Wang H, Xie X, Gu W, et al. Gene editing of economic macroalga Neopyropia yezoensis (Rhodophyta) will promote its development into a model species of marine algae[J]. New Phytologist, 2024, 244(5): 1687-1691.

Reviewer 2 Report
Comments and Suggestions for Authors
The work is mainly aimed to provide a comprehensive analyze of the C2H2 gene family of a representative intertidal seaweed species (Pyropia haitanensis) exploring it at genomic level and elucidating its biological functions.
The reason of the research is well placed, in fact, there is a limited number of studies regarding the role of C2H2 zinc-finger gene family in red algae (Rhodophyta)
The adopted technical approach is coherent with the research aims.
Results provide interesting insights about the role of C2H2 zinc-finger gene family in response to abiotic stress.
However, some revisions could improve the quality of the work:
- The abstract could be more concise, and summarize the most significant results
- In the “Introduction” section,
- Some interesting research works about the role of C2H2 zinc-finger gene family in plant stress response, such as that available at https://www.mdpi.com/2223-7747/13/24/3580,
https://bmcgenomics.biomedcentral.com/articles/10.1186/s12864-024-10273-7,
https://pmc.ncbi.nlm.nih.gov/articles/PMC9574186/ , could be mentioned
- In the “Results” section,
- Figure1, including subfigures, is not readable. I suggest to split it in different figures. Similarly for Figure3.
- in the “Conclusion” section
- I suggest to include some sentences about the impact in breeding programs of the achieved results
A careful rereading of the manuscript is strongly suggested. They are some oversights and inaccuracies:
- Please check the Italic form of genes – Lines 15, 16, 103, etc.
- Furthermore, scientific names when written for the first time must have the extended form – Lines 209, 210, etc.
Some minor issues:
- I suggest to remove the abstract subdivision in Introduction, Objectives, Methods, Results, Conclusion.
- Please check the reference format, https://www.mdpi.com/journal/ijms/instructions
Author Response
Dear Editors and Reviewer:
We would like to thank the editors and reviewer's work devoted to our manuscript and we are very grateful for your valuable suggestions.
According to the comments of the reviewers, we have corrected some errors. More unnecessary information has been cut out and more necessary information is stated. We have fully addressed each concern and hope that this revised manuscript is now acceptable. Each concern is discussed in detail below.
Reviewer 2Comment 1
In the “Introduction” section, some interesting research works about the role of C2H2 zinc-finger gene family in plant stress response, such as that available at
https://www.mdpi.com/2223-7747/13/24/3580,
https://bmcgenomics.biomedcentral.com/articles/10.1186/s12864-024-10273-7,
https://pmc.ncbi.nlm.nih.gov/articles/PMC9574186/, could be mentioned
Response 1: Thank you for your valuable suggestion. We have carefully reviewed the literature recommended by you and indeed provided us with more insights. We refer to these relevant results in the introduction to enrich the interpretation of the C2H2-ZFPs gene family on the corresponding stress in plants (in line 109-117).
Comment 2
In the “Results” section, Figure1, including subfigures, is not readable. I suggest to split it in different figures, similarly for Figure3.
Response 2: Thank you for your attention to this problem. According to your suggestion, we have split Figure 1 in the new manuscript and raised the resolution of Figure 3 to improve the readability of the article.
Comment 3
In the “Conclusion” section, I suggest to include some sentences about the impact in breeding programs of the achieved results.
Response 3: Thank you for the valuable comments, which certainly help us to improve the quality of our manuscript. In reference to your opinion, in the conclusion section, based on our research results, we discussed and prospected the contents related to the selection breeding and application of future stress-resistant strains.
Comment 4
A careful rereading of the manuscript is strongly suggested. They are some oversights and inaccuracies: Please check the Italic form of genes – Lines 15, 16, 103, etc. Furthermore, scientific names when written for the first time must have the extended form – Lines 209, 210, etc.
Response 4: Thank you for reminding us that this was indeed an oversight on our part. We have carefully reviewed the manuscript again and corrected the issues you mentioned.
Comment 5
The abstract could be more concise, and summarize the most significant results. I suggest to remove the abstract subdivision in Introduction, Objectives, Methods, Results, Conclusion.
Response 5: Thanks to your suggestions, we have revised the abstract to highlight important results and meet submission requirements.
Thanks again, the hard work and valuable suggestions of the editors and reviewers are of great help to our improvement in the manuscript. We sincerely hope that the revised article can meet the requirements of International journal of molecular science and be accepted.

Round 2
Reviewer 1 Report
Comments and Suggestions for Authors
The authors responded to all my inquiries.